# Significance of serum FGF-23 for risk assessment of contrast-associated acute kidney injury and clinical outcomes in patients undergoing coronary angiography

Shao-Sung Huang[1,2,3], Po-Hsun Huang[1,4,5]*, Hsin-Bang Leu[1,2,5], Tao-Cheng Wu[1,5], Jaw-Wen Chen[1,2,3], Shing-Jong Lin[1,2,5]

**1** Division of Cardiology, Department of Internal Medicine, Taipei Veterans General Hospital, Taipei, Taiwan, **2** Healthcare and Management Center, Taipei Veterans General Hospital, Taipei, Taiwan, **3** School of Medicine, National Yang Ming Chiao Tung University, Tainan City, Taiwan, **4** Department of Critical Care Medicine, Taipei Veterans General Hospital, Taipei, Taiwan, **5** Institute of Clinical Medicine, Cardiovascular Research Center, National Yang Ming Chiao Tung University, Tainan City, Taiwan

* huangbs@vghtpe.gov.tw

**Data Availability Statement:** Data cannot be shared publicly due to sensitive patient information. Data are available from the Taipei

## Abstract

### Background

Fibroblast growth factor (FGF)-23 levels rise as kidney function declines. Whether elevated FGF-23 levels are associated with an increased risk for contrast-associated acute kidney injury (CA-AKI) and major adverse cardiovascular events (MACE) in patients undergoing coronary angiography remain uncertain.

### Methods

In total, 492 patients receiving coronary angiography were enrolled. Their serum FGF-23 levels were measured before administration of contrast media. The occurrence of CA-AKI was defined as a rise in serum creatinine of 0.5 mg/dL or a 25% increase from the baseline value within 48 h after the procedure. All patients were followed up for at least 1 year or until the occurrence of MACE including death, nonfatal myocardial infarction (MI), and ischemic stroke.

### Results

Overall, CA-AKI occurred in 41 (8.3%) patients. During a median follow-up of 2.6 years, there were 24 deaths, 3 nonfatal MIs, and 7 ischemic strokes. Compared with those in the lowest FGF-23 tertile, individuals in the highest FGF-23 tertile had a significantly higher incidence of CA-AKI (P < 0.001) and lower incidence of MACE-free survival (P = 0.001). In multivariate regression analysis, higher FGF-23 level was found to be independently associated with a graded risk for CA-AKI (OR per doubling, 1.90; 95% CI 1.48–2.44) and MACE (HR per doubling, 1.25; 95% CI 1.02–1.52).

Veterans General Hospital Institutional Data Access / Ethics Committee (irbopinion@vghtpe.gov.tw) or via the corresponding author (huangbs@vghtpe. gov.tw) for researchers who meet the criteria for access to confidential data.

**Funding:** The author(s) received no specific funding for this work.

**Competing interests:** The authors have declared that no competing interests exist.

## Conclusions

Elevated FGF-23 levels were associated with an increased risk for CA-AKI and future MACE among patients undergoing coronary angiography. FGF-23 may play a role in early diagnosis of CA-AKI and predicting clinical outcomes after coronary angiography.

## Introduction

Contrast-induced nephropathy, also known as contrast-associated acute kidney injury (CA-AKI), remains a serious clinical problem associated with the use of iodinated contrast media in diagnostic imaging and interventional procedures [1, 2]. Several risk factors are associated with CA-AKI, including the pre-existence of renal dysfunction, hypotension, heart failure, diabetes mellitus, older age, anemia, and the amount and type of contrast media used [3–5]. Despite technological advances, the incidence of CA-AKI ranges from 1–2% in the general population to 50% in high-risk subgroups after coronary angiography or percutaneous coronary intervention (PCI) [6]. Although the exact pathophysiological mechanisms underlying CA-AKI are still unclear, multiple factors, including renal vasoconstriction, direct cytotoxic effects of contrast media, oxidative stress, inflammation, and tubular obstruction, are likely involved [7]. The development of CA-AKI after PCI has been associated with poor short- and long-term outcomes, including death resulting primarily from cardiovascular causes [8–10].

Fibroblast growth factor-23 (FGF-23) is a secreted, bone-derived hormone that plays a fundamental role in the regulation of phosphate and vitamin D homeostasis by stimulating urinary phosphate excretion and inhibiting the activation of vitamin D in the kidney [11, 12]. FGF-23 levels rise as renal function declines, and higher levels are strongly associated with increased risks of cardiovascular disease (CVD), heart failure, and progression to end-stage renal disease (ESRD) [13, 14]. In addition, elevated FGF-23 level has been shown to be an independent predictor of death in patients with chronic kidney disease (CKD) and ESRD [14, 15]. Recently, the association between FGF-23 levels and acute kidney injury (AKI) has also been studied. Patients with AKI show elevated FGF-23 levels, which are associated with a greater risk of death or need for renal replacement therapy [16]. Moreover, some clinical studies have described the relation between FGF-23 level and inflammatory markers. Mendoza et al. found that there is a positive association between levels of FGF-23 and IL-6, C-reactive protein (CRP), and TNF-α in patients with CKD [17]. Nasrallah et al. demonstrated that FGF-23 is strongly correlated with high-sensitivity CRP and advanced oxidation protein products in hemodialysis patients [18].

Given the role of oxidative stress and inflammation in the pathogenesis of CA-AKI and atherosclerosis, we hypothesized that elevated FGF-23 level is associated with a higher risk for CA-AKI and more cardiovascular events in patients undergoing coronary angiography. This study aimed to evaluate the relationship between serum FGF-23 levels and the incidence of CA-AKI, and to investigate the predictive role of FGF-23 in clinical outcomes of patients undergoing coronary angiography.

## Methods

### Study population

Between March 2011 and March 2015, a series of 492 consecutive patients who were admitted to a single medical center for elective coronary angiography were enrolled in this study. Before

enrollment, the medical record of each patient was reviewed in detail to collect data on medications, smoking status, and risk factors for CA-AKI such as age, pre-existing renal dysfunction, type 2 diabetes mellitus, and other comorbidities. Hypertension was defined as a systolic blood pressure $\geq$140 mmHg, a diastolic blood pressure $\geq$90 mmHg, or taking antihypertensive medications. Type 2 diabetes was defined as a fasting glucose $\geq$126 mg/dL or use of anti-hyperglycemic agents. CKD was defined as an estimated glomerular filtration rate (eGFR) <60 mL/min/1.73 m$^2$. eGFR was calculated using the modified glomerular filtration rate estimating equations for Chinese patients [19]. Body mass index (BMI) was calculated by dividing the weight of the patient in kilograms by the square of the height in meters. A history of CVD was established based on a self-reported history of heart failure, myocardial infarction (MI), or cerebrovascular accident. Patients were classified as former smokers only if they had not smoked for more than 6 months. Patients with a history or clinical evidence of renal failure during chronic peritoneal or hemodialytic treatment were excluded. Nonionic low-osmolar contrast media (iopromide) was administered intra-arterially for all patients, mainly through transradial catheters. Metformin and nephrotoxic agents such as NSAIDs were discontinued 48 hours prior to contrast media injection. Before and after contrast media exposure, isotonic (0.9%) saline was given intravenously at a rate of 1 mL/kg/h for 12 hours. In patients with reduced left ventricular function (ejection fraction <40%) or overt heart failure, the hydration rate was reduced to 0.5 mL/kg/h. All participants gave written informed consent, and the study protocol was approved by the institutional review board of Taipei Veterans General Hospital in Taipei, Taiwan.

## Laboratory investigations

After an overnight fast for $\geq$8 hours, blood samples were obtained from all patients before coronary angiography. Serum levels of CRP; glucose; calcium; phosphate; and lipid profiles including triglycerides, total cholesterol, high-density lipoprotein (HDL) cholesterol, and low-density lipoprotein (LDL) cholesterol were measured using a Hitachi 7600 Autoanalyzer (Hitachi Ltd., Tokyo, Japan). Serum creatinine concentration (SCr) was measured at the time of admission, and every day for the following 3 days after contrast media exposure. Serum intact FGF-23 levels were measured using enzyme-linked immunosorbent assay (Merck Millipore). All samples were measured in duplicate, with an intra-assay coefficient of variation <10%. The patients were classified into tertiles based on the distribution of FGF-23 in the study sample. Patients with FGF-23 levels >15.1 pg/mL (highest tertile) were defined as the high FGF-23 group (n = 164, 33.3%), those with FGF-23 levels <4.3 pg/mL (lowest tertile) were defined as the low FGF-23 group (n = 166, 33.7%) and those with FGF-23 levels $\geq$4.3 pg/mL and $\leq$15.1 pg/ml were defined as the intermediate FGF-23 group (n = 162, 32.9%).

## Clinical follow-up for end points

All patients were evaluated for the occurrence of CA-AKI, which was defined as the increment in SCr of 0.5 mg/dL or a 25% increase from the baseline value within 48 hours after coronary angiography. They were advised to visit the outpatient clinics regularly after discharge from the hospital. Besides, all patients were periodically contacted by telephone and their medical records were reviewed regularly until the occurrence of a major adverse kidney event (MAKE) which was a composite of death, the need for dialysis, or a persistent increase in SCr of at least 50% from baseline within the first 90 days after contrast media exposure, and a major adverse cardiovascular event (MACE) such as death, nonfatal MI, or ischemic stroke. Nonfatal MI was defined by elevated cardiac enzyme levels with ischemic symptoms and/or characteristic electrocardiographic changes. Ischemic stroke was defined as the presence of a new neurological

deficit lasting for at least 24 hours, with imaging evidence of a cerebrovascular accident confirmed by either computed tomography or magnetic resonance imaging.

## Statistical analysis

Data were expressed in terms of mean and standard deviation (SD) for numeric variables and as the number (percent) for categorical variables. Continuous variables between groups were compared using Student's *t*-test or one-way ANOVA. Subgroup comparisons of categorical variables were assessed using Chi-square or Fisher's exact test. Logarithmic (log) transformation was performed to achieve normal distribution for skewed variable (CRP). Survival curves were generated using the Kaplan–Meier method, and survival among the groups was compared using the log-rank test. Multivariate logistic regression analysis was used to investigate the relationship between serum FGF-23 levels and the incidence of CA-AKI. Multivariate Cox regression analysis was used to evaluate the association of FGF-23 with clinical outcomes. The primary analyses evaluated FGF-23 tertiles using the lowest tertile as the reference category. FGF-23 was also evaluated as a continuous variable after log base 2 transformation, interpreted as "per doubling." For CA-AKI, an initial model was adjusted for age and sex. A second model added hypertension, diabetes, prior heart failure, contrast volume, eGFR, serum CRP levels, and medications (loop diuretics and angiotensin-converting enzyme inhibitors/angiotensin receptor blockers). For MACE, an initial model was adjusted for age and sex. A second model added the risk factors for CVD, including hypertension, diabetes, smoking, prior stroke, prior MI, prior heart failure, serum CRP levels, CA-AKI, eGFR, and cardiovascular medications (angiotensin-converting enzyme inhibitors/angiotensin receptor blockers, β-blockers, and statins). The sample size was determined based on the need to detect the difference of incident CA-AKI between high and low FGF-23 groups with 80% power, using a cutoff for statistical significance of 0.05. Data were analyzed using SPSS version 17.0 (SPSS Inc, Chicago, IL). A *P* value <0.05 was considered to indicate statistical significance.

## Results

### Patient characteristics

The mean age of the 492 patients (331 males, 67%) was 68 ± 12 years. All patients were followed up until May 2016. None of them dropped out of the study, and all occurrences of adverse events were recorded. The baseline characteristics of the patients are shown in **Table 1**. There were no differences between the patients in the low FGF-23 group and those in the high FGF-23 group with respect to sex, BMI, smoking status, hypertension, serum levels of lipid profiles, fasting glucose and calcium, and contrast volume. However, compared with patients in the lowest FGF-23 tertile, those with higher FGF-23 levels were older, had a greater prevalence of type 2 diabetes and CKD, and were more likely to present with higher levels of serum phosphate and CRP at baseline. Individuals with higher FGF-23 levels also had a higher prevalence of prior heart failure, prior MI, and prior stroke/transient ischemic attack, and lower eGFR.

### Effects of FGF-23 levels on CA-AKI, MAKE and MACE

All patients were successfully followed up for a mean duration of 33.1 ± 14.5 months (median 31.5 months; range, 27–1890 days). Of these, 41 (8.3%) developed CA-AKI after coronary angiography, and 20 (4.1%) developed MAKE. In addition, a total of 34 MACEs occurred, including 24 deaths, 3 nonfatal MIs, and 7 ischemic strokes.

**Table 1. Baseline characteristics of the study population by tertiles of FGF-23.**

| | FGF-23 Tertile | | | |
| --- | --- | --- | --- | --- |
| | <4.3 pg/ml | 4.3–15.1 pg/ml | >15.1 pg/ml | |
| | (n = 166) | (n = 162) | (n = 164) | *P* |
| Age (years) | 66.9 ± 11.9 | 67.7 ± 12.1 | 70.3 ± 13.0 | 0.032 |
| Male | 116 (70) | 113 (70) | 102 (62) | 0.236 |
| Body mass index (kg/m2) | 25.6 ± 3.8 | 25.8 ± 4.3 | 25.5 ± 4.8 | 0.910 |
| Current smoker | 53 (32) | 56 (35) | 54 (33) | 0.877 |
| Hypertension | 105 (63) | 106 (65) | 114 (70) | 0.476 |
| Diabetes mellitus | 49 (30) | 48 (30) | 68 (42) | 0.031 |
| Chronic kidney disease | 34 (21) | 36 (22) | 78 (48) | <0.001 |
| Prevalent HF and CVD | | | | |
| History of HF | 12 (7) | 9 (6) | 25 (15) | 0.006 |
| History of MI | 4 (2) | 9 (6) | 14 (9) | 0.050 |
| History of stroke/TIA | 4 (2) | 8 (5) | 14 (9) | 0.044 |
| Lipid profiles (mg/dl) | | | | |
| Triglycerides | 120 ± 73 | 119 ± 69 | 139 ± 113 | 0.064 |
| Total cholesterol | 166 ± 32 | 160 ± 32 | 165 ± 38 | 0.199 |
| High-density lipoprotein | 44 ± 12 | 42 ± 11 | 41 ± 14 | 0.106 |
| Low-density lipoprotein | 101 ± 27 | 95 ± 28 | 98 ± 30 | 0.282 |
| Fasting glucose (mg/dl) | 103 ± 33 | 104 ± 29 | 109 ± 35 | 0.203 |
| Serum phosphate (mg/dl) | 3.3 ± 0.6 | 3.3 ± 0.6 | 3.6 ± 0.9 | 0.033 |
| Serum calcium (mg/dl) | 9.0 ± 0.6 | 8.9 ± 0.5 | 8.9 ± 0.7 | 0.240 |
| C-reactive protein (mg/dl) | 0.24 ± 0.28 | 0.47 ± 0.68 | 0.94 ± 1.06 | <0.001 |
| eGFR (ml/min/1.73m$^2$) | 74.5 ± 21.2 | 71.3 ± 18.2 | 62.2 ± 26.1 | <0.001 |
| Contrast volume (ml) | 100 ± 75 | 99 ± 72 | 97 ± 70 | 0.941 |

Values are mean ± SD or number (%).

HF: heart failure; CVD: cardiovascular disease; MI: myocardial infarction; TIA: transient ischemic attack; eGFR: estimated glomerular filtration rate.

Compared with patients without CA-AKI, those with CA-AKI had a higher prevalence of hypertension, type 2 diabetes, CKD, and heart failure, had lower eGFR, and were more likely to present with higher pre-procedural CRP levels. In addition, loop diuretics and angiotensin-converting enzyme inhibitors/angiotensin receptor blockers were used more frequently at baseline in patients with CA-AKI than in those without CA-AKI (shown in **S1 Table**). As illustrated in **Fig 1**, individuals in the highest FGF-23 tertile had a significantly higher incidence of CA-AKI than that in patients in the lowest FGF-23 tertile (17.7% vs. 2.4%, *P* < 0.001). In the high FGF-23 group, two of CA-AKI cases in which SCr doubled from baseline value. However, in the low and intermediate FGF-23 groups, none of CA-AKI cases in which SCr doubled from baseline value.

Moreover, there was a graded relationship of higher FGF-23 levels with higher event rates for each outcome except nonfatal MI, as shown in **Table 2**. The highest tertile of serum FGF-23 was strongly associated with higher risk of MAKE. Patients in the high FGF-23 group exhibited more cardiovascular deaths and all-cause mortality than those shown by patients in the low and intermediate FGF-23 groups. The incidence of MACE was significantly higher in patients with high FGF-23 levels than in those with low FGF-23 levels (12.2% vs. 2.4%, *P* = 0.002). In order to investigate the potential impact of baseline FGF-23 levels on adverse event-free survival, Kaplan–Meier survival analysis was performed. Patients in the highest

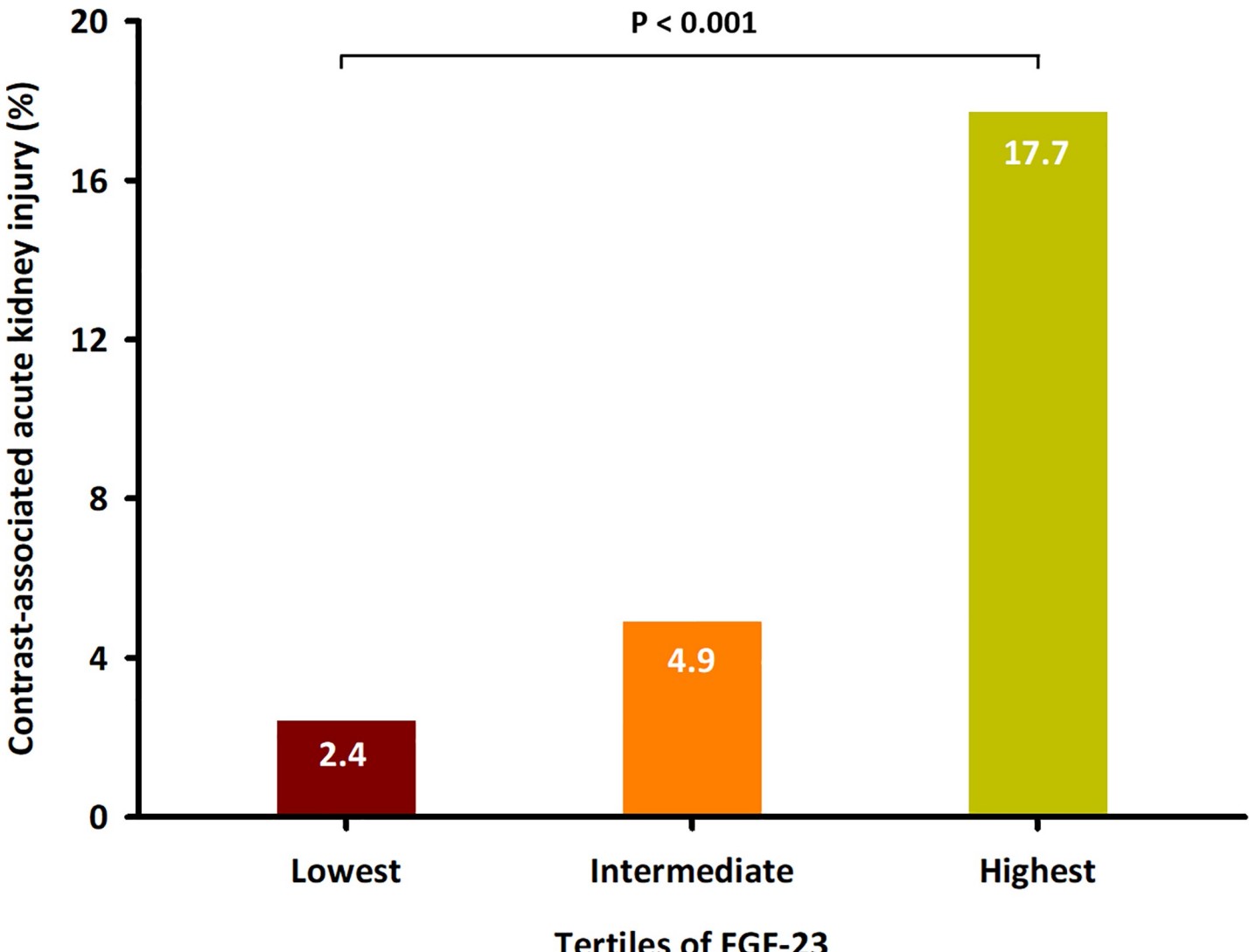

**Fig 1. Incidence of contrast-associated acute kidney injury in patients stratified into 3 groups according to tertiles of FGF-23.** $P < 0.001$.

**Table 2. Adverse events based on tertiles of serum FGF-23 levels.**

|  | FGF-23 Tertile | | |  |
| --- | --- | --- | --- | --- |
|  | <4.3 pg/ml | 4.3–15.1 pg/ml | >15.1 pg/ml |  |
|  | (n = 166) | (n = 162) | (n = 164) | *P* |
| MAKE | 2 (1.2) | 5 (3.1) | 13 (7.9) | 0.007 |
| All-cause death | 2 (1.2) | 6 (3.7) | 16 (9.8) | 0.001 |
| Cardiovascular death | 0 (0) | 3 (1.9) | 7 (4.3) | 0.023 |
| Nonfatal MI | 1 (0.6) | 2 (1.9) | 0 (0) | NS |
| Ischemic stroke | 1 (0.6) | 2 (1.2) | 4 (2.4) | NS |
| MACE (death/MI/stroke) | 4 (2.4) | 10 (6.2) | 20 (12.2) | 0.002 |

Values are number (%).

MAKE: major adverse kidney event; MI: myocardial infarction; MACE: major adverse cardiovascular event.

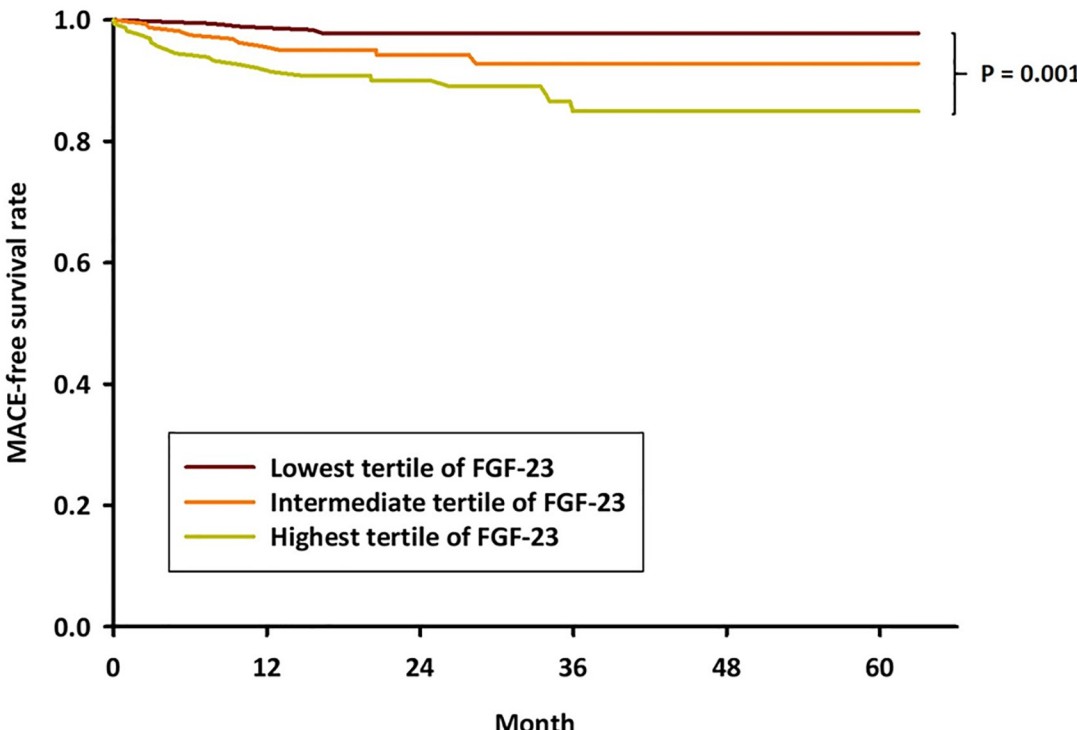

**Fig 2. Kaplan–Meier survival curves stratified by baseline FGF-23 levels.** The MACE-free survival rate was significantly reduced in patients with high FGF-23 levels ($P = 0.001$ by log-rank test).

FGF-23 tertile showed a significantly lower incidence of MACE-free survival than that shown by patients in the lowest FGF-23 tertile ($P = 0.001$), as illustrated in **Fig 2**.

## Independent correlates of CA-AKI and predictors of MACE

In order to investigate the association between FGF-23 and incident CA-AKI, multivariate logistic regression analysis was performed. Elevated FGF-23 levels remained significantly associated with a greater risk for CA-AKI after sequential multivariable adjustment, as shown in **Table 3**. Each doubling of FGF-23 levels was associated with a 103% higher risk in the primary model that adjusted for age and sex. The odds ratio (OR) comparing the highest versus the lowest tertile was 8.11 (95% confidence interval [CI], 2.76–23.82). In models that included other risk factors for CA-AKI, including hypertension, diabetes, prior heart failure, contrast volume, eGFR, log CRP, and medications (angiotensin-converting enzyme inhibitors/angiotensin receptor blockers and loop diuretics), the highest versus lowest tertile of FGF-23 was associated with a 6.82-fold greater risk for CA-AKI (95% CI, 2.22–21.01), and a 90% higher risk per doubling of FGF-23 levels.

In order to investigate the independent predictors of future MACE, multivariate Cox regression analysis was performed. In unadjusted Cox models, each doubling of FGF-23 levels was associated with a 40% increased risk of MACE, and ascending FGF-23 tertiles were associated with a stepwise increase in risk, as shown in **Table 3**. The hazard ratio (HR) comparing the highest versus the lowest tertile was 5.71 (95% CI, 1.95–16.73). The significant graded risk of MACE across the spectrum of FGF-23 levels persisted in all multivariable models. Each doubling of FGF-23 levels was associated with a 37% higher risk in the primary model that adjusted for age and sex. In the model that included other CVD risk factors, the highest versus

**Table 3. Risk of incident CA-AKI and incident MACE by baseline levels of FGF-23.**

| | FGF-23 Tertile | | | Linear model | |
|---|---|---|---|---|---|
| | <4.3 pg/ml | 4.3–15.1 pg/ml | >15.1 pg/ml | Per Doubling of FGF-23 | p* |
| OR (95% CI) of CA-AKI | | | | | |
| Unadjusted | 1.00 (ref) | 2.10 (0.62–7.13) | 8.70 (2.98–25.37) | 2.03 (1.61–2.56) | <0.001 |
| Age-sex-adjusted | 1.00 (ref) | 2.07 (0.61–7.03) | 8.11 (2.76–23.82) | 2.03 (1.60–2.57) | <0.001 |
| + CA-AKI risk factors† | 1.00 (ref) | 2.47 (0.70–8.70) | 6.82 (2.22–21.01) | 1.90 (1.48–2.44) | <0.001 |
| HR (95% CI) of MACE | | | | | |
| Unadjusted | 1.00 (ref) | 2.87 (0.90–9.17) | 5.71 (1.95–16.73) | 1.40 (1.18–1.66) | <0.001 |
| Age-sex-adjusted | 1.00 (ref) | 2.84 (0.89–9.07) | 5.31 (1.80–15.67) | 1.37 (1.16–1.63) | <0.001 |
| + CVD risk factors‡ | 1.00 (ref) | 2.72 (0.84–8.77) | 3.79 (1.20–11.99) | 1.25 (1.02–1.52) | 0.032 |

CA-AKI: contrast-associated acute kidney injury; MACE: major adverse cardiovascular event; CVD: cardiovascular disease.

*The p value for the linear logistic model.

†Adjusted for age, sex, hypertension, diabetes, prior heart failure, contrast volume, eGFR, log CRP, and medications (loop diuretics and angiotensin-converting enzyme inhibitors/angiotensin receptor blockers).

‡Adjusted for age, sex, smoking, prior stroke, prior myocardial infarction, prior heart failure, hypertension, diabetes, contrast-induced nephropathy, eGFR, log CRP, and medications (angiotensin-converting enzyme inhibitors/angiotensin receptor blockers, β-blockers, and statins).

lowest tertile of FGF-23 was associated with a 3.79-fold greater risk for MACE (95% CI, 1.20–11.99), and a 25% higher risk per doubling of FGF-23 levels.

## Discussion

The major finding of the present study is that elevated serum FGF-23 levels were independently associated with the incidence of CA-AKI and future cardiovascular events in patients undergoing coronary angiography. A graded relationship of higher FGF-23 levels with higher event rates for each outcome was observed. Elevated FGF-23 levels were more strongly associated with the risk for developing CA-AKI than MACE. Our findings did support the potential of FGF-23 as a novel risk marker for CA-AKI.

The widely usage of contrast media for a multitude of radiological procedures, including diagnostic coronary angiography and PCI, has raised concerns about the increasing incidence of a potential complication known as CA-AKI. However, the exact mechanism underlying CA-AKI is still unclear, although several suggestions have been put forward. Intrarenal vasoconstriction, inflammation, generation of reactive oxygen species, and direct tubular toxicity leading to hypoxia of the outer medulla are generally accepted as the main factors in the pathophysiology of CA-AKI [7]. A previous study has demonstrated that elevated CRP levels are associated with endothelial injury and impaired vasodilation, which may lead to acute renal damage and progressive loss of kidney function [20]. Moreover, systemic inflammation could render the kidneys more vulnerable to local inflammatory processes elicited by the reabsorption of iodinated contrast medium following angiographic procedures, favoring the development of CA-AKI [21–23]. Recent clinical studies indicate that there is a significant independent association between pre-procedural CRP levels and the occurrence of CA-AKI in patients with stable coronary artery disease undergoing elective PCI [24] and in those with ST-segment elevation MI undergoing primary PCI [25]. Consistent with the findings of previous studies, our data showed that elevated baseline CRP values are associated with higher incidence of CA-AKI in patients undergoing coronary angiography.

FGF-23, a phosphaturic peptide hormone secreted by the osteoblasts, is an important regulator of phosphorus and vitamin D metabolism [11, 12]. Accumulating evidence has linked the

components of phosphate homeostasis to inflammation and oxidative stress. Higher FGF-23 levels are independently associated with higher levels of inflammatory markers in patients with CKD and with substantially greater odds of manifesting severe inflammation [17]. FGF-23 is strongly correlated with various markers of inflammation and oxidative stress in ESRD patients on hemodialysis [18]. Moreover, elevation of plasma FGF-23 levels has been observed in multiple studies of human AKI [26]. Brown et el. reported that FGF-23 is independently associated with a higher risk of AKI hospitalizations in community-dwelling elderly individuals [27]. Neyra et al. demonstrated that serum intact FGF-23 levels were significantly higher in critically ill patients with AKI compared with matched controls without AKI. Elevated serum intact FGF-23 levels were associated with an increased risk of MAKE in critically ill patients admitted to the ICU [28]. In the current study, we also found that individuals with higher FGF-23 levels had higher baseline levels of inflammatory marker CRP, and that the risk of CA-AKI and MAKE were significantly increased in patients with elevated FGF-23 levels. Although one small study has shown that serum FGF-23 may have certain value in early diagnosis of CA-AKI [29], the causal relationship between FGF-23 and the development of CA-AKI has not been established. Our findings indicated an association between serum FGF-23 levels and the occurrence of CA-AKI, suggesting the predictive role of FGF-23 in the development of CA-AKI in patients undergoing coronary angiography. Although we were unable to determine whether FGF-23 has a causative effect, the potential mechanisms described above may partly explain the correlation between FGF-23 levels and the occurrence of CA-AKI following exposure to contrast media. Further studies are warranted to clarify the mechanistic role of FGF-23 in CA-AKI.

Multiple studies have demonstrated that elevated FGF-23 levels are associated with major cardiovascular events and mortality in patient with CKD and ESRD [14, 15, 30, 31]. Elevated FGF-23 levels are also strongly associated with adverse outcomes in patients with AKI. Leaf et al. reported that plasma FGF-23 levels rise early and predict AKI and death in patients undergoing cardiac surgery [32]. Moreover, higher FGF-23 levels are independently associated with greater mortality in critically ill patients. It may be a promising novel biomarker of AKI, death, and other adverse outcomes in critically ill patients [33, 34]. Recently, the association of FGF-23 with mortality is extended to individuals with heart failure with preserved ejection fraction [35], and to populations with known prevalent CVD [36, 37]. In addition, FGF-23 is independently associated with all-cause death and incident heart failure in community-living older individuals [13]. A previously published meta-analysis showed that individuals with increased plasma FGF-23 levels might suffer a higher risk of all-cause mortality and cardiovascular mortality [38]. Our findings also indicated the role of FGF-23 in predicting long-term outcomes in patients with suspected coronary artery disease. Consistent with the results of previous studies, our results showed significantly increased incidence of MACE, mainly death, in patients with high FGF-23 levels than in those with low FGF-23 levels. Further, serum FGF-23 levels could be independently associated with the risk of future MACE among patients undergoing coronary angiography.

This study had some limitations that should be considered. First, the study population was relatively small, and all participants were of Asian ethnicity and were recruited from a single center. Further studies with a larger number of different participants are required to confirm our findings. Second, the measurements of mineral metabolism complementary to FGF-23 (serum calcitriol and klotho) were not available. Thus, we could not provide additional insights into another potential mechanism underlying the association between elevated FGF-23 levels and CA-AKI. Third, the weight of the composite outcomes (MAKE, MACE) is mostly driven by mortality. Finally, our results are observational in nature, so residual confounding may

persist, and the causal effects of FGF-23 cannot be determined. Further studies are needed to clarify the exact interaction between FGF-23 and CA-AKI.

In conclusion, FGF-23 is likely signaling a worse health status in the FGF-23 upper-level group, since this group presented significantly more comorbidities (DM, CKD, previous MI and stroke) and more importantly, higher inflammation (higher CRP) and especially lower eGFR. Therefore, it is very likely that this worse health status was actually responsible for the results: greater CA-AKI and worse MACE. Nevertheless, our results show promise for testing this biomarker as part of novel risk prediction models of renal and cardiovascular outcomes in patients undergoing coronary angiography.

## Supporting information

**S1 Table. Baseline characteristics of patients with or without CA-AKI.**
(DOCX)

## Author Contributions

**Supervision:** Hsin-Bang Leu, Tao-Cheng Wu, Jaw-Wen Chen, Shing-Jong Lin.

**Writing – original draft:** Shao-Sung Huang.

**Writing – review & editing:** Po-Hsun Huang.

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
