## [Decision Letter · Decision Letter 0]

4 May 2021

PONE-D-21-02755

Significance of serum FGF-23 for risk assessment of contrast-induced nephropathy and clinical outcomes in patients undergoing coronary angiography

PLOS ONE

Dear Dr. Huang,

Thank you for submitting your manuscript to PLOS ONE. After careful consideration, we feel that it has merit but does not fully meet PLOS ONE’s publication criteria as it currently stands. Therefore, we invite you to submit a revised version of the manuscript that addresses all the points raised during the review process by the two reviewers.

We look forward to receiving your revised manuscript.

Kind regards,

Emmanuel A Burdmann

Academic Editor

PLOS ONE

Journal Requirements:

2) We note that you have included the phrase “data not shown” in your manuscript. Unfortunately, this does not meet our data sharing requirements. PLOS does not permit references to inaccessible data. We require that authors provide all relevant data within the paper, Supporting Information files, or in an acceptable, public repository. Please add a citation to support this phrase or upload the data that corresponds with these findings to a stable repository (such as Figshare or Dryad) and provide and URLs, DOIs, or accession numbers that may be used to access these data. Or, if the data are not a core part of the research being presented in your study, we ask that you remove the phrase that refers to these data.

3) PLOS requires an ORCID iD for the corresponding author in Editorial Manager on papers submitted after December 6th, 2016. Please ensure that you have an ORCID iD and that it is validated in Editorial Manager. To do this, go to ‘Update my Information’ (in the upper left-hand corner of the main menu), and click on the Fetch/Validate link next to the ORCID field. This will take you to the ORCID site and allow you to create a new iD or authenticate a pre-existing iD in Editorial Manager. Please see the following video for instructions on linking an ORCID iD to your Editorial Manager account: https://www.youtube.com/watch?v=_xcclfuvtxQ

4)  Please provide additional details regarding participant consent. In the ethics statement in the Methods and online submission information, please ensure that you have specified (1) whether consent was informed and (2) what type you obtained (for instance, written or verbal, and if verbal, how it was documented and witnessed). If your study included minors, state whether you obtained consent from parents or guardians. If the need for consent was waived by the ethics committee, please include this information.

5)  Thank you for submitting the above manuscript to PLOS ONE. During our internal evaluation of the manuscript, we found some minor occurrences of overlapping text with the following previous publication(s), some of which you are an author, which needs to be addressed:

- https://journals.lww.com/jhypertension/Abstract/2013/11000/Association_of_central_pulse_pressure_with.15.aspx

- https://jasn.asnjournals.org/content/25/2/349.full

- https://www.ahajournals.org/doi/10.1161/JAHA.117.008157

We would like to make you aware that copying extracts from previous publications word-for-word, especially outside the methods section, is unacceptable. In addition, the reproduction of text from published reports has implications for the copyright that may apply to the publications.

Please revise the manuscript to quote or rephrase the duplicated text and cite your sources for text outside the methods section. Please note that further consideration is dependent on the submission of a manuscript that addresses these concerns about the overlap in text with published work.

Reviewers' comments:

Reviewer's Responses to Questions

**Comments to the Author**

1. Is the manuscript technically sound, and do the data support the conclusions?

Reviewer #1: Yes

Reviewer #2: Yes

2. Has the statistical analysis been performed appropriately and rigorously? 

Reviewer #1: Yes

Reviewer #2: Yes

3. Have the authors made all data underlying the findings in their manuscript fully available?

Reviewer #1: Yes

Reviewer #2: Yes

4. Is the manuscript presented in an intelligible fashion and written in standard English?

Reviewer #1: Yes

Reviewer #2: Yes

5. Review Comments to the Author

Reviewer #1: The authors present their study "Significance of serum FGF-23 for risk assessment of contrast-induced nephropathy

and clinical outcomes in patients undergoing coronary angiography". I congratulate the authors for this work which highlights the value of FGF-23 in the risk-classification of AKI, in particular CA-AKI. Here are my comments to improve this report:

1-The authors should use a more contemporary term such as contrast-associated AKI (CA-AKI) rather than contrast-induced nephropathy (CIN) as it has been recently adopted by Radiology and Nephrology groups (PMID: 31961246), particularly when the cause of AKI cannot be attributed with 100% certainty

2-One limitation to acknowledge is the low incidence of CIN/CA-AKI (8.3%) and the lack of severity classification. Therefore, I suggest the authors to report the number of CIN/CA-AKI cases in which SCr doubled from baseline value according to tertiles of FGF-23 (if any). Further, major adverse kidney events (MAKE) could be also reported if data were collected (e.g., death, dialysis dependence, eGFR drop >30% from baseline or SCr >=50% above baseline). The latter outcome could be explored within the first 90 days of contrast exposure and also reported according to tertiles of FGF-23. This is a more relevant clinical outcome with very low incidence (<5%) according to the recent PRESERVE trial

3-Reporting MACE is a little misleading given the low number of cases of CV death, ischemic stroke and non-fatal MI. It is clear that the weight of the association is mostly driven by all-cause death. I suggest the authors to report all-cause death as the main dependent variable in the models

4-It is not clear if the authors measured intact vs. C-term FGF-23 levels. Please clarify

5-Consider citing other studies highlighting the association of FGF-23 levels with AKI and MAKE (PMID: 32123869)

6-Please specify if the coronary angiography procedures were all elective or if emergent cases were also included

7-Please specify the duration of isotonic saline infusion before the exposure to contrast

8-Please add to the Methods text the medications included in the multivariable models

9-The discussion of FGF-23 data in CKD is too long, perhaps focusing on the interaction between FGF-23, Klotho and vitD in AKI could be more informative for the reader, specially because the authors highlight the lack of these measures as a limitation of the study -although no background information was provided in the Discussion

Reviewer #2: To the authors,

The manuscript by Shao-Sung et al describes an interesting association between FGF-23 levels and CIN and MACE in patients submitted to coronary angiography. Despite the multivariate analysis, which included most comorbidites, findings are mainly observational and a causal relationship between FGF-23 and CIN and outcomes could not be demonstrated. Nevertheless, a significant association was clearly found. Accordingly, I suggest that in the causal relationship discussion, authors should include that FGF-23 is likely signalling a worse health status in the FGF-23 upper level group, since this group presented significantly more comorbidities (DM, CKD, previous MI and stroke) an more importantly, higher inflammation (higher PCR) and especially lower eGFR. Therefore, it is very likely that this worse health status was actually responsible for the results: greater CIN and worse MACE. For this reason, I suggest that the statement, in both conclusions (abstract and main text), about reduction or modulation of FGF-23 could prevent CIN should be removed since a causal relationship was not proven. Instead, authors should include the MACE analysis together with CIN results in the conclusions.

Minor points:

Figures - level of significance should be pointed or demonstrated between which groups (upper vs. lower)

Figure 1 - correct intermidiate title group (Intermediate)

6. PLOS authors have the option to publish the peer review history of their article (what does this mean?). If published, this will include your full peer review and any attached files.

Reviewer #1: **Yes: **Javier Neyra

Reviewer #2: **Yes: **Luis Yu

---

## [Author Response · Author response to Decision Letter 0]

17 Jun 2021

Response to Comments by Reviewer #1

Thank you very much for your interest in our paper and for your most instructive comments. We have revised the manuscript on the basis of your suggestions. The responses to your comments are as follow.

Comments to the Author

The authors present their study "Significance of serum FGF-23 for risk assessment of contrast-induced nephropathy and clinical outcomes in patients undergoing coronary angiography". I congratulate the authors for this work which highlights the value of FGF-23 in the risk-classification of AKI, in particular CA-AKI.

Here are my comments to improve this report:

1. The authors should use a more contemporary term such as contrast-associated AKI (CA-AKI) rather than contrast-induced nephropathy (CIN) as it has been recently adopted by Radiology and Nephrology groups (PMID: 31961246), particularly when the cause of AKI cannot be attributed with 100% certainty.

Response:

Thank you for your suggestions. We appreciate your recommendation that the term “contrast-associated acute kidney injury (CA-AKI)” should be more precise. Thus, we have made the appropriate revisions in the manuscript.

Contrast-induced nephropathy, also known as contrast-associated acute kidney injury (CA-AKI), remains a serious clinical problem associated with the use of iodinated contrast media in diagnostic imaging and interventional procedures. (Please see page 4/line 2–3)

2. One limitation to acknowledge is the low incidence of CIN/CA-AKI (8.3%) and the lack of severity classification. Therefore, I suggest the authors to report the number of CIN/CA-AKI cases in which SCr doubled from baseline value according to tertiles of FGF-23 (if any). Further, major adverse kidney events (MAKE) could be also reported if data were collected (e.g., death, dialysis dependence, eGFR drop >30% from baseline or SCr >=50% above baseline). The latter outcome could be explored within the first 90 days of contrast exposure and also reported according to tertiles of FGF-23. This is a more relevant clinical outcome with very low incidence (<5%) according to the recent PRESERVE trial.

Response:

Thank you for your suggestions and recommendations. As you recommended, we have made the appropriate revisions in the manuscript.

(1) Methods:

Besides, all patients were periodically contacted by telephone and their medical records were reviewed regularly until the occurrence of a major adverse kidney event (MAKE) which was a composite of death, the need for dialysis, or a persistent increase in SCr of at least 50% from baseline within the first 90 days after contrast media exposure, and a major adverse cardiovascular event (MACE) such as death, nonfatal MI, or ischemic stroke. (Please see page 8/line 13–15)

(2) Results:

Of these, 41 (8.3%) developed CA-AKI after coronary angiography, and 20 (4.1%) developed MAKE. (Please see page 11/line 10)

In the high FGF-23 group, two of CA-AKI cases in which SCr doubled from baseline value. However, in the low and intermediate FGF-23 groups, none of CA-AKI cases in which SCr doubled from baseline value. (Please see page 11/line 19 & page 12/line 1-3)

Moreover, there was a graded relationship of higher FGF-23 levels with higher event rates for each outcome except nonfatal MI, as shown in Table 2. The highest tertile of serum FGF-23 was strongly associated with higher risk of MAKE. (Please see page 12/line 5-6)

3. Reporting MACE is a little misleading given the low number of cases of CV death, ischemic stroke and non-fatal MI. It is clear that the weight of the association is mostly driven by all-cause death. I suggest the authors to report all-cause death as the main dependent variable in the models.

Response:

Thank you for your suggestions and recommendations. We agree with you that the number of cases of cardiovascular death, ischemic stroke and nonfatal MI is relatively low, and the weight of the association is mostly driven by all-cause death. However, the term MACE is arguably the most commonly used composite end point in cardiovascular research. In the current study, MACE was defined as a composite of all-cause death, nonfatal MI, or ischemic stroke. Because our study was undertaken to investigate the predictive role of FGF-23 in clinical outcomes of patients undergoing coronary angiography, we reported the association of FGF-23 with the risk of MACE. Our data showed that the incidence of MACE was significantly higher in patients with high FGF-23 levels than in those with low FGF-23 levels. Serum FGF-23 levels could be independently associated with the risk of future MACE among patients undergoing coronary angiography.

4. It is not clear if the authors measured intact vs. C-term FGF-23 levels. Please clarify.

Response:

Thank you for your suggestions and recommendations. As you recommended, we have made the appropriate revisions in the manuscript.

Serum intact FGF-23 levels were measured using enzyme-linked immunosorbent assay (Merck Millipore). (Please see page 7/line 16)

5. Consider citing other studies highlighting the association of FGF-23 levels with AKI and MAKE (PMID: 32123869)

Response:

Thank you for your suggestions and recommendations. As you recommended, we have made the appropriate revisions in the manuscript.

(1) Discussion:

Neyra et al. demonstrated that serum intact FGF-23 levels were significantly higher in critically ill patients with AKI compared with matched controls without AKI. Elevated serum intact FGF-23 levels were associated with an increased risk of MAKE in critically ill patients admitted to the ICU. (Please see page 16/line 2–5)

(2) References:

Neyra JA, Li X, Mescia F, Ortiz-Soriano V, Adams-Huet B, Pastor J, et al. Urine Klotho Is Lower in Critically Ill Patients With Versus Without Acute Kidney Injury and Associates With Major Adverse Kidney Events. Crit Care Explor. 2019;1:e0016. (Please see page 23/reference 28)

6. Please specify if the coronary angiography procedures were all elective or if emergent cases were also included.

Response:

Thank you for your suggestions and recommendations. As you recommended, we have made the appropriate revisions in the manuscript.

Between March 2011 and March 2015, a series of 492 consecutive patients who were admitted to a single medical center for elective coronary angiography were enrolled in this study. (Please see page 6/line 3)

7. Please specify the duration of isotonic saline infusion before the exposure to contrast.

Response:

Thank you for your suggestions and recommendations. As you recommended, we have made the appropriate revisions in the manuscript.

Before and after contrast media exposure, isotonic (0.9%) saline was given intravenously at a rate of 1 mL/kg/h for 12 hours. (Please see page 7/line 2–3)

8. Please add to the Methods text the medications included in the multivariable models.

Response:

Thank you for your suggestions and recommendations. As you recommended, we have made the appropriate revisions in the manuscript.

Methods:

For CA-AKI, an initial model was adjusted for age and sex. A second model added hypertension, diabetes, prior heart failure, contrast volume, eGFR, serum CRP levels, and medications (loop diuretics and angiotensin-converting enzyme inhibitors/angiotensin receptor blockers). For MACE, an initial model was adjusted for age and sex. A second model added the risk factors for CVD, including hypertension, diabetes, smoking, prior stroke, prior MI, prior heart failure, serum CRP levels, CA-AKI, eGFR, and cardiovascular medications (angiotensin-converting enzyme inhibitors/angiotensin receptor blockers, β-blockers, and statins). (Please see page 9/line 19 & page 10/line 1, 4–5)

9. The discussion of FGF-23 data in CKD is too long, perhaps focusing on the interaction between FGF-23, Klotho and vitD in AKI could be more informative for the reader, especially because the authors highlight the lack of these measures as a limitation of the study -although no background information was provided in the Discussion.

Response:

Thank you for your suggestions and recommendations. As you recommended, we have made the appropriate revisions in the manuscript.

Discussion:

Moreover, elevation of plasma FGF-23 levels has been observed in multiple studies of human AKI. Brown et el. reported that FGF-23 is independently associated with a higher risk of AKI hospitalizations in community-dwelling elderly individuals. Neyra et al. demonstrated that serum intact FGF-23 levels were significantly higher in critically ill patients with AKI compared with matched controls without AKI. Elevated serum intact FGF-23 levels were associated with an increased risk of MAKE in critically ill patients admitted to the ICU. (Please see page 15/line 18–19 & page 16/line 1–5)

Multiple studies have demonstrated that elevated FGF-23 levels are associated with major cardiovascular events and mortality in patient with CKD and ESRD. Elevated FGF-23 levels are also strongly associated with adverse outcomes in patients with AKI. Leaf et al. reported that plasma FGF-23 levels rise early and predict AKI and death in patients undergoing cardiac surgery. Moreover, higher FGF-23 levels are independently associated with greater mortality in critically ill patients. It may be a promising novel biomarker of AKI, death, and other adverse outcomes in critically ill patients. (Please see page 17/line 1–6)

Response to Comments by Reviewer #2

Thank you very much for your interest in our paper and for your most instructive comments. We have revised the manuscript on the basis of your suggestions. The responses to your comments are as follow.

Comments to the Author

1. The manuscript by Shao-Sung et al describes an interesting association between FGF-23 levels and CIN and MACE in patients submitted to coronary angiography. Despite the multivariate analysis, which included most comorbidites, findings are mainly observational and a causal relationship between FGF-23 and CIN and outcomes could not be demonstrated. Nevertheless, a significant association was clearly found. Accordingly, I suggest that in the causal relationship discussion, authors should include that FGF-23 is likely signaling a worse health status in the FGF-23 upper-level group, since this group presented significantly more comorbidities (DM, CKD, previous MI and stroke) and more importantly, higher inflammation (higher CRP) and especially lower eGFR. Therefore, it is very likely that this worse health status was actually responsible for the results: greater CIN and worse MACE. For this reason, I suggest that the statement, in both conclusions (abstract and main text), about reduction or modulation of FGF-23 could prevent CIN should be removed since a causal relationship was not proven. Instead, authors should include the MACE analysis together with CIN results in the conclusions.

Response:

Thank you for your suggestions and recommendations. As you recommended, we have made the appropriate revisions in the manuscript.

(1) Abstract:

Elevated FGF-23 levels were associated with an increased risk for CA-AKI and future MACE in patients undergoing coronary angiography. FGF-23 may play a role in early diagnosis of CA-AKI and predicting clinical outcomes after coronary angiography. (Please see page 3/line 1–4)

(2) Discussion:

In conclusion, FGF-23 is likely signaling a worse health status in the FGF-23 upper-level group, since this group presented significantly more comorbidities (DM, CKD, previous MI and stroke) and more importantly, higher inflammation (higher CRP) and especially lower eGFR. Therefore, it is very likely that this worse health status was actually responsible for the results: greater CA-AKI and worse MACE. Nevertheless, our results show promise for testing this biomarker as part of novel risk prediction models of renal and cardiovascular outcomes in patients undergoing coronary angiography. (Please see page 18/line 11–18)

Note: Because using a more contemporary term such as contrast-associated acute kidney injury (CA-AKI) rather than contrast-induced nephropathy (CIN) was recommended by Reviewer #1, we have made the appropriate revision in the manuscript.

Contrast-induced nephropathy, also known as contrast-associated acute kidney injury (CA-AKI), remains a serious clinical problem associated with the use of iodinated contrast media in diagnostic imaging and interventional procedures. (Please see page 4/line 2–3)

2. Figures - level of significance should be pointed or demonstrated between which groups (upper vs. lower)

Response:

Thank you for your suggestions and recommendations. As you recommended, we have made the appropriate revisions in the manuscript. (Please see Figure 1 & Figure 2)

3. Figure 1 - correct intermidiate title group (Intermediate)

Response:

Thank you for your suggestions and recommendations. As you recommended, we have made the appropriate revisions in the manuscript. (Please see Figure 1)

Thank you very much again and we appreciate your comments!

---

## [Decision Letter · Decision Letter 1]

6 Jul 2021

Significance of serum FGF-23 for risk assessment of contrast-associated acute kidney injury and clinical outcomes in patients undergoing coronary angiography

PONE-D-21-02755R1

Dear Dr. Huang,

We’re pleased to inform you that your manuscript has been judged scientifically suitable for publication and will be formally accepted for publication once it meets all outstanding technical requirements.

Please, note that Reviewer 1 recommend the authors to briefly acknowledge in the Limitations that the weight of the composite outcomes (MAKE, MACE) is mostly driven by mortality. This will further guide the reader in the interpretation of the results.

Kind regards,

Emmanuel A Burdmann

Section Editor

PLOS ONE

Additional Editor Comments (optional):

Reviewers' comments:

Reviewer's Responses to Questions

**Comments to the Author**

1. If the authors have adequately addressed your comments raised in a previous round of review and you feel that this manuscript is now acceptable for publication, you may indicate that here to bypass the “Comments to the Author” section, enter your conflict of interest statement in the “Confidential to Editor” section, and submit your "Accept" recommendation.

Reviewer #1: All comments have been addressed

Reviewer #2: All comments have been addressed

2. Is the manuscript technically sound, and do the data support the conclusions?

Reviewer #1: Yes

Reviewer #2: Yes

3. Has the statistical analysis been performed appropriately and rigorously? 

Reviewer #1: Yes

Reviewer #2: Yes

4. Have the authors made all data underlying the findings in their manuscript fully available?

Reviewer #1: Yes

Reviewer #2: Yes

5. Is the manuscript presented in an intelligible fashion and written in standard English?

Reviewer #1: Yes

Reviewer #2: Yes

6. Review Comments to the Author

Reviewer #1: The authors have addressed this reviewer's comments and suggestions and have significantly improved their manuscript. I recommend the authors to briefly acknowledge in the Limitations that the weight of the composite outcomes (MAKE, MACE) is mostly driven by mortality. This will further guide the reader in the interpretation of the results.

Reviewer #2: (No Response)

7. PLOS authors have the option to publish the peer review history of their article (what does this mean?). If published, this will include your full peer review and any attached files.

Reviewer #1: **Yes: **Javier Neyra

Reviewer #2: No

---

## [Editor Report · Acceptance letter]

16 Jul 2021

PONE-D-21-02755R1 

Significance of serum FGF-23 for risk assessment of contrast-associated acute kidney injury and clinical outcomes in patients undergoing coronary angiography 

Dear Dr. Huang:

I'm pleased to inform you that your manuscript has been deemed suitable for publication in PLOS ONE. Congratulations! Your manuscript is now with our production department. 

Kind regards, 

on behalf of

Dr. Emmanuel A Burdmann 

Section Editor

PLOS ONE